# Tumor Progression Reverses Cardiac Hypertrophy and Fibrosis in a Tetracycline-Regulated ATF3 Transgenic Mouse Model

**DOI:** 10.3390/cells12182289

**Published:** 2023-09-15

**Authors:** Lama Awwad, Ami Aronheim

**Affiliations:** Department of Cell Biology and Cancer Science, Ruth and Bruce Rappaport Faculty of Medicine, Technion—Israel Institute of Technology, Haifa P.O. Box 9649, Israel; lamaaw@campus.technion.ac.il

**Keywords:** cardiac dysfunction, hypertrophy, fibrosis, transgenic mice, cancer, tumor progression, immune system, macrophages

## Abstract

Cardiovascular diseases (CVD) and cancer are the top deadly diseases in the world. Both CVD and cancer have common risk factors; therefore, with the advances in treatment and life span, both diseases may occur simultaneously in patients. It is becoming evident that CVD and cancer are highly connected, establishing a novel discipline known as cardio-oncology. This includes the cardiomyocyte death following any anti-tumor therapy known as cardiotoxicity as well the intricate interplay between heart failure and cancer. Recent studies, using various mouse models, showed that heart failure promotes tumor growth and metastasis spread. Indeed, patients with heart failure were found to be at higher risk of developing malignant diseases. While the effect of heart failure on cancer is well established, little is known regarding the effect of tumors on heart failure. A recent study from our lab has demonstrated that tumor growth and metastasis ameliorate cardiac remodeling in a pressure-overload mouse model. Nevertheless, this study was inconclusive regarding whether tumor growth solely suppresses cardiac remodeling or is able to reverse existing heart failure outcomes as well. Here, we used a regulable transgenic mouse model for cardiac hypertrophy and fibrosis. Cancer cell implantation suppressed cardiac dysfunction and fibrosis as shown using echocardiography, qRT-PCR and fibrosis staining. In addition, tumor growth resulted in an M1 to M2 macrophage switch, which is correlated with cardiac repair. Macrophage depletion using clodronate liposomes completely abrogated the tumors’ beneficial effect. This study highly suggests that harnessing tumor paradigms may lead to the development of novel therapeutic strategies for CVDs and fibrosis.

## 1. Introduction

Heart failure (HF) and cancer are two of the most studied diseases worldwide, due to their high mortality rates [1,2]. Recently, many studies have demonstrated an intimate link between cardiac dysfunction and tumor progression [3,4]. It is well documented that HF in mice that was induced by various methods including myocardial infraction (MI) [5,6], pressure overload [7] a transgenic model for cardiac hypertrophy and fibrosis [8] and hypertension mouse model [9], promotes the progression of different types of cancer including intestinal adenoma [5], breast cancer [6,7,8] and lung carcinoma [6,7,8]. Furthermore, it was shown that patients with HF after MI have an increased risk of developing cancer [7,10,11]. A recently published study has showed that early cardiac remodeling processes, prior to HF, are sufficient to promote cancer cell proliferation and metastasis seeding [7]. The phenotype was explained by the involvement of the immune system and the secretion of multiple cytokines that induce cancer cell proliferation [3,4]. While the effect of HF on cancer progression is well defined, less is known about the opposite direction, namely, tumors’ effect on cardiac dysfunction. A case in point is a recently published study that demonstrated that cancer cell growth of either breast or lung cancer ameliorates cardiac dysfunction and suppresses remodeling processes following pressure overload induced by Transverse Aortic Constriction (TAC) surgery [12]. Tumor-bearing mice showed an enhanced cardiac contractility function, and a decrease in cardiac hypertrophy and fibrosis levels. This phenotype was shown to be mediated by the involvement of the innate immune system and more specifically an M2 macrophage switch in the heart [12,13]. The TAC model involves a major surgical intervention that may have side effects. In addition, since cancer cells were implanted prior to pressure overload intervention, it was impossible to distinguish whether cancer cells solely suppressed the cardiac remodeling or they are also able to reverse existing cardiac dysfunction outcomes. Towards this end, we utilized a transgenic mouse model that overexpresses the transcriptional factor ATF3 in a doxycycline-regulable manner (tet-off system) [14]. ATF3 is an immediate early gene expressed under multiple stress conditions and growth stimuli, including cardiac events [15,16]. Moreover, patients with HF display high levels of the ATF3 protein [17,18]. ATF3 overexpression in cardiomyocytes is sufficient to induce heart hypertrophy and cardiac fibrosis, resulting in cardiac contractile dysfunction [19,20]. In order to examine whether cancer cells are able to reverse existing heart failure outcomes, we implanted cancer cells into mice pre-conditioned with heart hypertrophy and fibrosis. To prevent deterioration of the cardiac condition in the course of cancer growth, ATF3 expression was simultaneously transcriptionally suppressed by the addition of doxycycline to the diet, thus arresting cardiac remodeling any further.

In this study, we show that tumor-bearing ATF3 transgenic mice display decreased cardiac hypertrophy and fibrosis levels compared to tumor-free mice. The amelioration of the cardiac outcome was dependent on an M1 to M2 macrophage switch. Collectively, the data suggest that cancer cells are able to both suppress and reverse heart failure outcomes. Harnessing cancer paradigms may provide novel strategies for the treatment of chronic heart failure, which are direly needed.

## 2. Materials and Methods

All experimental protocols were approved by the Institutional Committee for Animal Care and Use at the Technion, Israel Institute of Technology, Faculty of Medicine, Haifa, Israel (approval number IL-157-10–21). All study procedures complied with the guidelines from the NIH Guide for the Care and Use of Laboratory Animals.

### 2.1. Animals

All mice used were backcrossed to a C57Bl/6 background for over 6 generations. The ATF3 transgenic mouse model is the result of mating of two transgenic mice: the first expresses the human influenza hemagglutinin (HA) fused to the human ATF3 under the control of the tetracycline activator (tTA) regulatory DNA elements. The second expresses the tTA transcription factor under the control of the αMHC promoter (αMHC-tTA), directing tTA expression to cardiomyocytes [14]. The tTA protein binding to the promoter is regulated by doxycycline (tet-off system). Double transgenic mice containing both the αMHC-tTA and ATF3 transgenes express the human ATF3 in cardiomyocytes and are hereafter designated ATF3 transgenic mice.

Breeding cages include mice harboring either the ATF3 transgene [20] or αMHC-tTA [21]. Mating cages were given regular chow supplemented with doxycycline (0.2 mg/mL, Sigma D9891, St. Louis, MO, USA) in the drinking water containing 5% sucrose to counteract the bitter taste of the antibiotic. Mouse genotyping was performed as described previously [20]. Upon weaning at three weeks of age, regular water was provided to allow ATF3 transgene expression in cardiomyocytes. Hearts, tumors and lungs were collected for further analysis. The number of mice used in each experiment is indicated in the figure legends.

### 2.2. Cell Culture

The Lewis Lung Carcinoma (LLC) cell line was purchased from the American Type Culture Collection (ATCC) [22]. The LLC cell line was tested and found to be free of mycoplasma and viral contamination. The cells were cultured in DMEM containing 10% FBS, 1% streptomycin and penicillin, 1% L-glutamine and 1% sodium pyruvate at 37 °C in a humidified atmosphere containing 5% CO_2_.

### 2.3. Cancer Implantation

LLC cells (10^6^ cells per mouse) were orthotopically injected subcutaneously into the flanks. Tumor size was measured using a caliper, and tumor volume was calculated with the formula: Width^2^ × Length × 0.5. The humane endpoint was defined as when the tumor size reaches a maximum volume of 1500 mm^3^, according to the Institutional Animal Care and Use Committee.

### 2.4. Echocardiography

Mice were anesthetized with 1% isoflurane and kept on a 37 °C heated plate throughout the procedure. Echocardiography was performed with a Vevo3100 micro-ultrasound imaging system (VisualSonics, Fujifilm, Tokyo, Japan) equipped with a 22- to 55-MHz (MS550D) linear array transducer. Cardiac size, shape and function were analyzed using conventional 2-dimensional imaging and M-mode recordings. Maximal left ventricular end-diastolic (LVDd) and end-systolic (LVDs) dimensions were measured in short-axis M-mode images. Fractional shortening (FS) was calculated with the following formula: FS% = [(LVDd − LVDs)/LVIDd] × 100. The FS value was based on the average of at least 3 separate measurements for each mouse.

### 2.5. Fibrosis Staining

Heart or tumor tissue was fixed in 4% formaldehyde overnight, embedded in paraffin, serially sectioned at 10 μm intervals, and then mounted on slides. Masson’s trichrome staining was performed according to the standard protocol. Images were acquired using 3DHistech Panoramic 250 Flash III (3DHISTECH Ltd., Budapest, Hungary). Each section was fully scanned. The percent of the interstitial fibrosis was determined as the ratio of the fibrosis area to the total area of the heart section using ImageJ software.

### 2.6. Cell Size Staining

Heart tissue was fixed in 4% formaldehyde overnight, embedded in paraffin, serially sectioned at 10 µm intervals and then mounted on slides. Sections were stained following deparaffinization with FITC-conjugated wheat germ agglutinin (Sigma Aldrich Cat #L4895) and diluted to a 1:30 with phosphate-buffered saline (PBS). The sections were washed three times with PBS and mounted in Fluorescence Mounting Medium (Dako, S3023, Santa Clara, CA, USA). Images were acquired using 3DHistech Panoramic 250 Flash III (3DHISTECH Ltd., Budapest, Hungary). Cell size was analyzed by calculating the cross sectional area using Image J software.

### 2.7. RNA Extraction

RNA was extracted from hearts and tumor using an Aurum total RNA fatty or fibrous tissue kit (No. 732-6830, Bio-Rad, Hercules, CA, USA) according to the manufacturer’s instructions. Next, cDNA was synthesized from 1000 ng purified mRNA using the iScript cDNA Synthesis Kit (No. 170–8891, Bio-Rad).

### 2.8. qRT-PCR

Quantitative real-time polymerase chain (qRT-PCR) was performed with QuantStudio3 (Thermofisher Scientific, 5823 Newton Drive, Carlsbad, CA, USA). Values were normalized to the Hsp90 housekeeping gene, as indicated in the figure legends. All oligonucleotide sequences used are found in Appendix A. The results are presented as fold change, compared to control mice.

### 2.9. Heart Single-Cell Suspension and Flow Cytometry

Heart single-cell suspension and flow cytometry were prepared as previously described [23]. Briefly, hearts were perfused, extracted and finely minced. Then, they were incubated with digestive enzymes at 37 °C on a rocking shaker at 50 rpm for 45–60 min. The samples were homogenized with a 40 µm cell strainer. Red blood cells were lysed using Ammonium–Chloride–Potassium (ACK) lysis buffer. The samples were centrifuged at 400× *g* for 5 min at 4 °C and the pellet was resuspended with FACS buffer. The cells were immuno-stained with the following anti-mouse antibodies: CD45- Alexa Fluor^®^ 700 (BioLegend, 103128, San Diego, California, USA), CD11b-PerCP-Cy 5.5 (BioLegend, 101228), F480-PE (BioLegend, 123110) and CD206-BV421 (BioLegend, 141717). The cells were incubated (30 min, 4 °C) with the antibody mixture in staining buffer and analyzed using an LSRFortessa flow cytometer (BD Biosciences). The data were analyzed using FlowJo V.10 software (FlowJo, Ashland, OR, USA).

### 2.10. Macrophage Depletion

Macrophage depletion was performed using either clodronate- or saline-containing liposomes (Encapsula Nanosciences LLS, Brentwood, TN, USA, SKU# CLD-8901). Liposomes were intraperitoneally (IP) injected twice a week until the experimental endpoint according to the manufacturer’s instructions.

### 2.11. Statistics

The data are presented as mean ± SEM. All mice were included in each statistical analysis unless they were euthanized for humane reasons before the experimental endpoint. Animals were chosen in a randomized fashion for each experimental group. Statistical significance was determined by comparing the two means using Student *t*-test. Statistical analysis and outlier exclusion were performed using GraphPad Prism 8 software (La Jolla, CA, USA). Values of *p* < 0.05 were accepted as statistically significant.

## 3. Results

### 3.1. LLC Tumor Growth Ameliorates Cardiac Dysfunction in ATF3 Transgenic Mice

To examine how tumor growth affects cardiac dysfunction, we used an ATF3 transgenic mouse model expressing the transcription factor ATF3 under the control of a tetracycline promoter [20]. To direct ATF3 expression to cardiomyocytes, mice were crossed with αMHC-tetracycline-activator transgenic line to produce double transgenic mice. The mice were kept on a doxycycline diet until weaning to prevent developmental defects. Upon weaning and doxycycline removal, ATF3 is expressed in cardiomyocytes leading to heart hypertrophy, fibrosis and cardiac dysfunction [20] and cancer progression [8]. To arrest ATF3-dependent cardiac deterioration, doxycycline was added four weeks after weaning followed by cancer cell implantation. Two groups of transgenic mice were included: one group was subcutaneously injected with LLC cancer cells and the other group was left un-injected (Figure 1A). Cardiac function was assessed by echocardiography prior to sacrifice and fractional shortening (FS) was calculated. Tumor-free double transgenic mice displayed a relatively low FS (23.79 ± 2.02) and cardiac function was not significantly deteriorated upon Dox addition (Figure 1B and Appendix A). In contrast, tumor-bearing mice that showed decreased FS levels prior to cancer cells injection, displayed improved FS (29.06 ± 1.71) post injection (Figure 1B,C). In the latter, FS reached cardiac contractile levels that are similar to those of healthy mice (30%, Figure 1B,C) [24]. This indicated an enhanced cardiac contractility function following tumor implantation. The mice were sacrificed and the ventricle weight to body weight ratio (VW/BW) was calculated (Figure 1D). Tumor-bearing mice showed a lower VW/BW ratio indicating reduced hypertrophy, which was confirmed by a reduced cardiomyocytes cross sectional area as seen by hemagglutinin staining of sections derived from tumor-bearing mice compared with sections derived from tumor-free mice (Figure 1E). This reduced hypertrophy in tumor-bearing mice was also accompanied by a statistically significant reduction in hypertrophic hallmark gene marker ANP and a reduction trend in BNP expression (Figure 1F,G).

Next, we sought to examine the tumors’ effect on cardiac fibrosis, another hallmark of cardiac remodeling and heart failure seen in the ATF3 transgenic mouse model. Thus, heart sections were stained for fibrosis using Masson’s trichrome staining (Figure 2A). Consistently, quantitative analysis of fibrosis deposition showed a decrease in fibrosis levels in the tumor-bearing mice (Figure 2B). Similarly, a decrease in the expression of fibrosis hallmark gene markers tumor growth factor (TGFβ3) and connective tissue growth factor (CTGF) accompanied the decrease in cardiac fibrosis in the tumor-bearing mice (Figure 2C,D).

### 3.2. Tumor-Bearing Mice Exhibit M1 to M2 Macrophage Polarization in the Heart

Macrophages are known to play a crucial role in inflammation, which is another hallmark of maladaptive cardiac remodeling. Cardiac macrophages are subdivided into two main categories: M1 inflammatory macrophages, which are known to worsen the cardiac remodeling state, and M2 anti-inflammatory macrophages that are important for cardiac tissue repair. To examine how the cardiac macrophage populations are affected by the presence of a solid tumor, we analyzed cardiac immune cells using flow cytometry and qRT-PCR. Flow cytometry analysis revealed an elevation of the anti-inflammatory M2 cardiac macrophages using the F4/80 + CD206+ markers in tumor-bearing mice compared to tumor-free ATF3 tg mice (Figure 3A). These data were also supported by qRT-PCR analysis, using hallmark gene markers for general macrophages (F4/80; Figure 3B), M2 macrophages (CD163 and CD206; Figure 3C,D), and pro-inflammatory M1 macrophages (TNFα; Figure 3E). Moreover, qRT-PCR analysis showed higher levels of the cytokine CCL2 in the hearts of tumor-bearing mice (Figure 3F). CCL2 is known to play an important role in inducing macrophage polarization toward the anti-inflammatory state.

Collectively, these data show a potential role of cardiac M2 macrophages in mediating the beneficial effect in tumor-bearing ATF3 transgenic mice.

### 3.3. Macrophage Depletion Abrogates Tumor-Dependent Amelioration of Cardiac Dysfunction

To examine the role of macrophages in mediating the tumor-dependent cardiac phenotype, we used liposomes containing either clodronate (to deplete mouse macrophage populations) or saline-containing liposomes as control. During the course of the experiment, liposomes were injected twice a week starting at the same time as cancer cell implantation (Figure 4A). While the mice receiving the control injected liposomes showed a higher FS following tumor implantation, tumor-bearing clodronate-injected mice showed no improvement in FS, namely, in the absence of macrophages, cancer-dependent cardiac contractility improvement was completely abrogated (Figure 4B). Consistently, following sacrifice, the clodronate-treated groups exhibited no difference in the VW/BW ratio (Figure 4C). Flow cytometry and qRT-PCR analysis confirmed the complete depletion of cardiac macrophage populations in the clodronate-treated group compared to the saline group (Appendix A).

Collectively, these results indicate that macrophages in general, and specifically M2 macrophages, play a key role in regulating the beneficial effect of tumors on cardiac dysfunction in ATF3 tg mice.

## 4. Discussion

Cardiovascular-associated diseases are the number one killer, contributing to one in four deaths in the USA [25]. Drug therapies to treat CVD and HF typically reduce volume over-load and heart rate without directly targeting the molecular defects within the heart. The number of novel cardiovascular drugs developed in the past 20 years is very limited and their efficacy and long-term toxicity are of concern [26]. Fibrosis, which is typically associated with HF especially following myocardial infarction (MI), is part of wound healing process characterized by injury, inflammation, myofibroblast activation, matrix deposition and remodeling [27]. The pathophysiologic principles of fibrosis are shared by several diseases involving the liver, heart, skeletal muscle, kidney, and lung [28,29,30,31]. Currently, there is no good treatment for fibrotic diseases and fibrotic lesions are considered irreversible [32]. Therefore, heart failure patients suffer from chronic exhaustion and reduced life quality with a five-year overall survival [25]. During recent efforts from our laboratory to explore the interplay between heart failure and cancer, we were surprised to identify low cardiac remodeling parameters in TAC-operated tumor-bearing mice [7]. However, since this finding was anecdotal and not the main purpose of our original study which aimed to uncover the effect of early cardiac remodeling on cancer progression, this finding required further thorough investigation. This led to the provocative finding that tumor growth ameliorates cardiac dysfunction [12]. Another study using an MI model performed detailed cardiac analysis in the presence and absence of tumors and failed to observe any overt differences in the heart between the experimental groups [6]. Whether differences in the severity of the models or the use of different breast cancer cell lines explain the inconsistent findings has yet to be determined. Nevertheless, we next performed a series of experiments addressing the role of tumor progression on cardiac dysfunction in TAC-operated mice [12]. Our findings are consistent with the notion that tumor growth suppresses cardiac hypertrophy, reduces fibrosis deposition and improves cardiac dysfunction. Yet, since we implanted cancer cells prior to TAC operation, we could not conclude whether the cancer cells are able reverse the cardiac remodeling outcome. In an additional study from our lab, we used a MDX mouse model for Duchenne Muscular Dystrophy. These mice suffer from reduced cardiac contractile function with no apparent hypertrophy. In addition, these mice display skeletal and diaphragm muscle fibrosis. Cancer cell implantation in 4–6-month-old MDX mice that display massive fibrosis significantly reduced fibrosis deposition [33], demonstrating the possible role of tumors in reversing existing fibrotic lesions [34]. Here, we used the regulable ATF3 mouse model to study cardiac remodeling and fibrosis [8,20]. The role of ATF3 in cardiac hypertrophy is a matter of debate. On the one hand, ATF3 deficiency was reported to result in cardiac hypertrophy and fibrosis. On the other hand, we showed that ATF3 KO only has a minor effect on cardiac remodeling [24]. Yet, in the absence of ATF3 and the Jun dimerization protein 2 (JDP2), another AP-1 repressor protein, mice are protected from pressure overload-induced cardiac remodeling [24]. Significantly, mice with cardiomyocyte ATF3 expression develop cardiac hypertrophy and fibrosis [8,19,20]. In the tetracycline-regulable transgenic mouse model, it is possible to induce adult cardiac remodeling following ATF3 expression. This allowed us to examine the possibility that tumor growth can reverse the hallmarks of the cardiac remodeling phenotype including ventricle hypertrophy, cardiomyocyte hypertrophy and fibrosis. Indeed, tumor-bearing mice displayed reduced heart failure outcomes compared to the non-tumor-bearing control group. Cardiac contractile function (FS) reached 30%, which is similar to the FS of naïve mice [20]. In addition, in the transgenic model and consistent with the TAC model, the macrophage populations are a crucial component. The fact that macrophage depletion completely abrogated this process strongly highlights the importance of harnessing innate immune cells for therapeutic purposes. Although it was shown that the anti-inflammatory M2 macrophages are able to repair cardiac function [35], no treatment has been developed for CVD. Although cancer cell implantation is not considered a plausible treatment, understanding the mechanism underlying M1 to M2 polarization balance may serve as a novel therapeutic strategy. Further studies in this direction are currently ongoing in our laboratory to mimic tumors’ beneficial effect on the heart in the absence of a growing tumor.

Limitations: The M1 and M2 macrophages were identified based on the expression pattern of a set of markers that are transmembrane glycoproteins, enzymes, growth factors, hormones, cytokines and cytokine receptors. Although these markers are frequently used in this research field**,** some of them can be expressed on other cell types and thus may not be unique to macrophage populations.

## 5. Conclusions

Using a spontaneous cardiac hypertrophy and fibrosis mouse model, we showed that cancer cell implantation followed by tumor progression resulted in a consistent amelioration of cardiac dysfunction, and reduced hypertrophy and fibrosis. This was accompanied by macrophage M1 to M2 polarization in the heart and this effect was completely abrogated in the absence of macrophages. These findings suggest that the reversion of cardiac dysfunction, hypertrophy and fibrosis is possible. Thus, novel therapeutics strategies may be developed based on these original and provocative findings.

## Figures and Tables

**Figure 1 cells-12-02289-f001:**
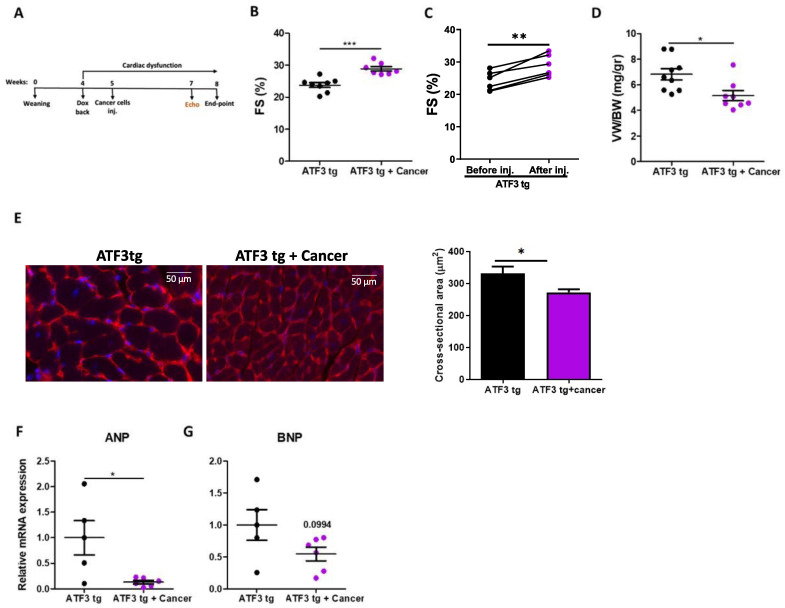
LLC tumors have a beneficial effect on cardiac contractility and hypertrophy in ATF3 tg male mice. (**A**) Experimental timeline. Two groups of ATF3 tg male mice were included in the experiment. All mice were weaned at the age of 4 weeks (week 0); 4 weeks later, doxycycline was removed. A week later, the mice were either injected with LLC cancer cells (*n* = 7) or left untreated (*n* = 8). (**B**) Echocardiography was performed prior to sacrifice and the fractional shortening percent (FS%) was calculated according to the parameters in Appendix A. (**C**) FS% of tumor-bearing ATF3 tg mice before and after cancer injection. (**D**) The ventricular/body weight ratio (VW/BW). (**E**) Heart sections (*n* = 4) were stained with wheat germ hemagglutinin and cell size was analyzed (5 sections per heart). Scale bar = 50 μm. The levels of hypertrophic gene markers ANP (**F**) and BNP (**G**) were measured using qRT-PCR using cDNA derived from heart mRNA of either tumor-free (*n* = 5) or tumor-bearing ATF3 tg mice (*n* = 6) and normalized to the Hsp90 housekeeping gene. Results are presented as mean ± SEM; Student’s *t*-test was performed. * *p* < 0.05, ** *p* < 0.01, *** *p* < 0.001.

**Figure 2 cells-12-02289-f002:**
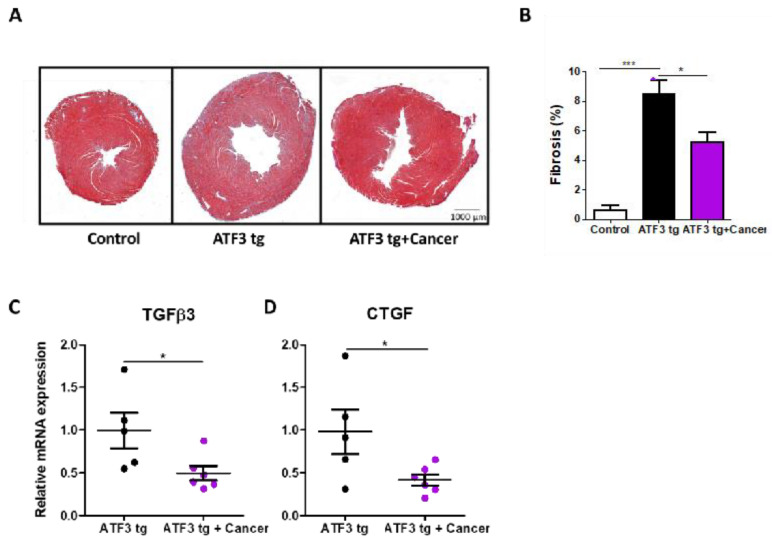
LLC tumors ameliorate cardiac fibrosis in ATF3 tg mice. (**A**) Representative heart sections of control, ATF3 tg and tumor-bearing ATF3 tg mice (*n* = 4 each) embedded in paraffin and stained with Masson’s trichrome stain to visualize fibrosis. Scale bar = 1000 μm. (**B**) Quantification of cardiac fibrosis levels (%). Fibrosis hallmark genes TGFβ3 (**C**) and CTGF (**D**) were measured using qRT-PCR using mRNA derived from heart mRNA of either ATF3 tg or tumor-bearing ATF3 tg mice and normalized to the Hsp90 housekeeping gene. Results are presented as mean ± SEM; Student’s *t*-test was performed. * *p* < 0.05, *** *p* < 0.001.

**Figure 3 cells-12-02289-f003:**
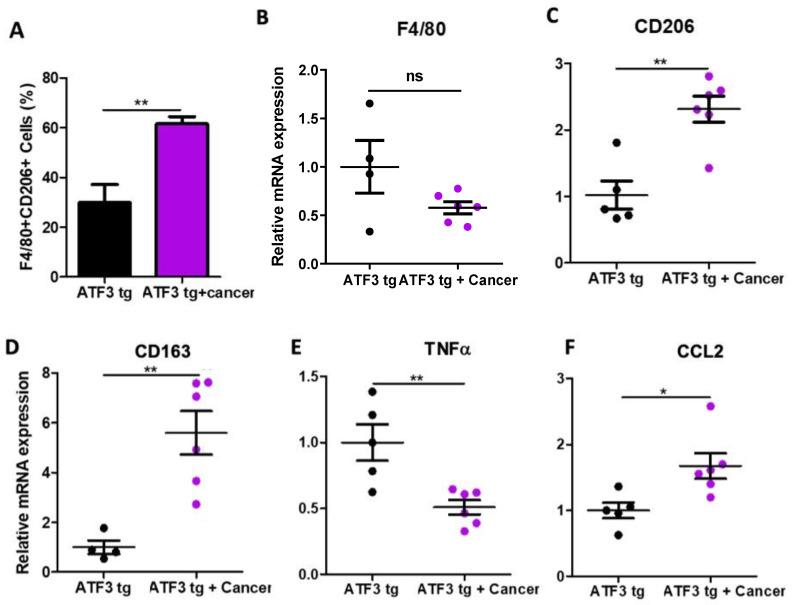
Tumor-bearing ATF3 tg mice exhibit M1 to M2 macrophage polarization in the heart. (**A**) Flow cytometry analysis of cardiac M2 macrophages (F4/80 + CD206+) in either tumor-free or tumor-bearing ATF3 tg mice (*n* = 4 per group). Macrophage marker F4/80. Gating strategy can be found in Appendix A. (**B**), M2 macrophage markers CD206 and CD163 (**C**,**D**), M1 macrophage marker TNFα (**E**) and M2-polarizing cytokine CCL2 (**F**) levels were measured by qRT-PCR using cDNA derived from heart mRNA of either ATF3 tg or tumor-bearing ATF3 tg mice and normalized to the Hsp90 housekeeping gene. Results are presented as mean ± SEM; Student’s *t*-test was performed. * *p* < 0.05, ** *p* < 0.01, ns is for non-significant changes.

**Figure 4 cells-12-02289-f004:**
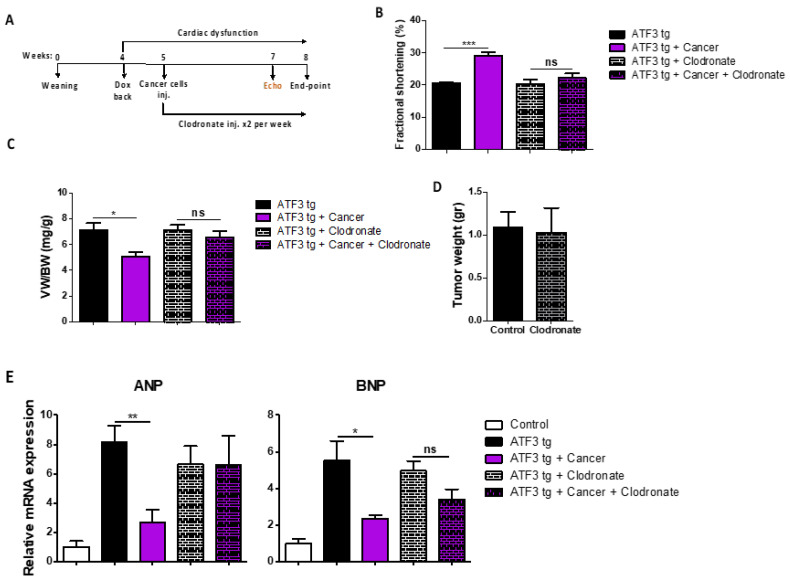
Cardiac macrophage depletion results in the loss of the tumor-dependent beneficial effect on cardiac function of ATF3 tg mice. (**A**) Experimental timeline. Four groups were included in this experiment: ATF3 tg and tumor-bearing ATF3 tg mice injected with either saline- or clodronate-containing liposomes (*n* = 5 each). (**B**) Echocardiography was performed prior to sacrifice and the fractional shortening percent (FS%) was calculated according to the parameters in Appendix A. (**C**) The ventricular/body weight ratio (VW/BW). (**D**) Tumor weight (gr) of saline- or clodronate-containing injected groups. (**E**) Hypertrophic hallmark genes ANP and BNP levels were measured by qRT-PCR using mRNA derived from heart mRNA of the experimental groups and normalized to the Hsp90 housekeeping gene. Results are presented as relative to non-transgenic mice indicated as control. Mean ± SEM; one-way ANOVA followed by Tukey post-test (**B**,**C**,**E**) or Student’s *t*-test (**D**) were performed. * *p* < 0.05, ** *p* < 0.01, *** *p* < 0.001. Non-significant differences are indicated as “ns”.

## Data Availability

Data and materials are available upon request to the corresponding author.

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
