# Peer review of "Tumor Progression Reverses Cardiac Hypertrophy and Fibrosis in a Tetracycline-Regulated ATF3 Transgenic Mouse Model"

_cells, 2023, doi:10.3390/cells12182289_

Round 1
Reviewer 1 Report (Previous Reviewer 1)
The revised manuscript by Awwad and Aronheim has improved significantly, but there are still some issues that need to be addressed before publication:
1. Line 24: typing error “… beneficial eff”
2. Line 71: typing error “directly”
3. Lines 144-155: different font size and style.
4. Line 192/Figure 1G: BNP is still not significantly reduced. While the authors claim to have highlighted this in the revised manuscript, the statement that both ANP and BNP are reduced needs to be corrected (lines 192-194).
5. On this point, I wondered about the discrepancy between Figure 1F-G and Figure 4E. In the latter, BNP is significantly reduced (column 2 vs. 3), which is not the case in Figure 1. Also, there is no statement about Figure 4D-F in the results section.
6. Line 198: typing error “…(week 0)…” and “… Doxycycline was removed.”
7. Line 199: typing error “A week after, mice…”
8. The upper part of Figure 4 (A-D) is in poor quality.
9. Figure 1E: Scale bar is missing in the left image.
10. Lines 283-287: different font size and style.
In addition, previous items 14 and 16 were not answered. Please address these points:
14. The conclusion from Figure 2B in line 205, that the tumor-group reaches fibrosis levels of the control group is not correct. A look on the data from Figure 2B shows a 5-fold higher level of fibrotic tissue in the “ATF3 tg + Cancer” group. Please also provide the p-value for this comparison in the figure.
16. Figure 3A.: Please provide p-value for F4/80. Is there a reduction in total macrophage number by cancer induction?
The English language is generally okay, but can be improved.
Author Response
We wish to thank the reviewer for finding our manuscript interesting and suitable for publication.
Point to point response:
The revised manuscript by Awwad and Aronheim has improved significantly, but there are still some issues that need to be addressed before publication:
- Line 24: typing error “… beneficial eff”
Response: Corrected in text.
- Line 71: typing error “directly”
Response: It’s not a typing error. “Direly” means strongly needed.
- Lines 144-155: different font size and style.
Response: Corrected in text.
- Line 192/Figure 1G: BNP is still not significantly reduced. While the authors claim to have highlighted this in the revised manuscript, the statement that both ANP and BNP are reduced needs to be corrected (lines 192-194).
Response: Corrected in text.
- On this point, I wondered about the discrepancy between Figure 1F-G and Figure 4E. In the latter, BNP is significantly reduced (column 2 vs. 3), which is not the case in Figure 1. Also, there is no statement about Figure 4D-F in the results section.
Response: Figure 1 and Figure 4 are separate experiments. In Figure 1, BNP is not statistically significant, although there is a reduction trend in tumor-bearing mice.
- Line 198: typing error “…(week 0)…” and “… Doxycycline was removed.”
Response: Corrected in text.
- Line 199: typing error “A week after, mice…”
Response: Corrected in text.
- The upper part of Figure 4 (A-D) is in poor quality.
Response: The quality of Figure 4 was edited.
- Figure 1E: Scale bar is missing in the left image.
Response: Scale bar was added.
- Lines 283-287: different font size and style.
Response: Font size was corrected accordingly.
In addition, previous items 14 and 16 were not answered. Please address these points:
- The conclusion from Figure 2B in line 205, that the tumor-group reaches fibrosis levels of the control group is not correct. A look on the data from Figure 2B shows a 5-fold higher level of fibrotic tissue in the “ATF3 tg + Cancer” group. Please also provide the p-value for this comparison in the figure.
Response: Sentence was corrected, P-value was added.
- Figure 3A.: Please provide p-value for F4/80. Is there a reduction in total macrophage number by cancer induction?
Response: Not statistically significant. P-value is 0.1016.
Comments on the Quality of English Language
The English language is generally okay, but can be improved.
Response: The manuscript has been proofread and English quality was also revised.
Reviewer 2 Report (Previous Reviewer 3)
Specific comments:
The details added to the legends and methods make it easier to understand the design of the experiments. Nevertheless, there are still some points that need clarification. In particular, I still don't understand how all the experiments were carried out on the number of mice included in the groups? 8 and 7 respectively indicated in legend fig 1 but there are 9 and 8 points in graph 1D. In addition, 4 hearts were cut for labelling and 5 and 6 for RNA preparation, which would indicate 9 and 10 animals per group. Finally, it is not indicated how many animals were used for cytometry, which would mean at least 3 more mice per group. Can the authors clarify this?
In methods, line 166, the authors talk about "outlier exclusion": which points and in which experiments are these points excluded? In particular, are all the parameters of a mouse excluded, or single results from several mice, and why?
In the results: Line 175-176: the authors present previously published results on which they base their current work. It would be preferable to use the results of this previous work in the introduction or methods in order to justify the choices made for this study
Line 185-187: the authors compare their results with results on control mice from another study. This is a discussion, this remark should be moved
Figure 1C: the result of the statistics must be indicated
Figure 1E: Cellular hypertrophy in ATF3Tg mice is known; however, the new result is that cancer reduces this hypertrophy. As this result was obtained on 5 slices per heart and 4 hearts, it is easy to quantify this by counting the number of cells per field or by measuring the size of the cells with membrane labelling. Statistics could be applied to this major functional result. In addition, it is possible to use the ultrasound results to demonstrate reversion of cardiac hypertrophy ? this would reinforce the functional results
Figure 2A and B: A CTL group appears but it is still not described in the methods or in the experimental timeline Fig1A? Why and what are these mice? Are these data from previous articles published and cited by the authors, ref 20, 8, 24?
Figure 2B: statistics between CTL and ATF3tg+cancer must be indicated
Line 212: “quantitative analysis of fibrosis deposition showed a decrease in fibrosis levels in the tumor‐bearing mice group that almost reached the control levels (Figure 2B).” the level of fibrosis seems to be very different between the CTL and ATF3tg+cancer groups, and statistics are missing
Figure 3A: the number of mice used for cytometry must be indicated in the figure legend
Figure 4: the quality of this figure is poor and there is a problem with the title
Figure 4E: The CTL group is added in this qPCR analysis. Why only in this one and not in all the other qPCR analyses? and from which mice do these results come?
Minor points :
Line 24: text problem in the added sentence
Line 123: tumours are not used in this article
Line 144-155: font size different from the rest of the article
Line 280-281 : I don't understand the meaning of this sentence? Comment made previously but no response. Can the authors specify what they mean by "peripheral symptoms"?
The limitations paragraph must be moved up before the general conclusion
There is still not enough information to understand supplementary figure 1 and the cytometry results are still illegible because they are very small, fig 2 and 3
This version needs to be proofread, as some sentences lack punctuation, have text errors, space problems, etc.
Author Response
We wish thank the reviewer for the valuable comments that significantly improved our manuscript and made it more understandable.
point to point response:
Specific comments:
The details added to the legends and methods make it easier to understand the design of the experiments. Nevertheless, there are still some points that need clarification. In particular, I still don't understand how all the experiments were carried out on the number of mice included in the groups? 8 and 7 respectively indicated in legend fig 1 but there are 9 and 8 points in graph 1D. In addition, 4 hearts were cut for labelling and 5 and 6 for RNA preparation, which would indicate 9 and 10 animals per group. Finally, it is not indicated how many animals were used for cytometry, which would mean at least 3 more mice per group. Can the authors clarify this?
In methods, line 166, the authors talk about "outlier exclusion": which points and in which experiments are these points excluded? In particular, are all the parameters of a mouse excluded, or single results from several mice, and why?
Response: In the experiment described in figure 1, exclusion was performed only on the RNA analysis level due to low quality of RNA. All other results from these mice were included in the analysis.
For staining and labeling, only 4 hearts were randomly selected.
In the results: Line 175-176: the authors present previously published results on which they base their current work. It would be preferable to use the results of this previous work in the introduction or methods in order to justify the choices made for this study
Response: The transgenic mice which we base the current study on, are described in methods, Animals section (Line 80-94).
Line 185-187: the authors compare their results with results on control mice from another study. This is a discussion, this remark should be moved
Response: The sentence describing the FS of control mice was left for clarity of control mice comparison. A sentence about this result was added in the discussion.
Figure 1C: the result of the statistics must be indicated
Response: Statistics have been added.
Figure 1E: Cellular hypertrophy in ATF3Tg mice is known; however, the new result is that cancer reduces this hypertrophy. As this result was obtained on 5 slices per heart and 4 hearts, it is easy to quantify this by counting the number of cells per field or by measuring the size of the cells with membrane labelling. Statistics could be applied to this major functional result. In addition, it is possible to use the ultrasound results to demonstrate reversion of cardiac hypertrophy ? this would reinforce the functional results
Response: Cell size measuring was added. Additional ultrasound results were added to supplementary material. Ultrasound results besides FS are presented in the supplementary table 2 and 3. The FS parameter is calculated based on LVID;d and LVID;s.
Figure 2A and B: A CTL group appears but it is still not described in the methods or in the experimental timeline Fig1A? Why and what are these mice? Are these data from previous articles published and cited by the authors, ref 20, 8, 24?
Response: These mice are littermates single transgenic (healthy) that were used as control for fibrosis staining.
Figure 2B: statistics between CTL and ATF3tg+cancer must be indicated
Response: Statistics were added.
Line 212: “quantitative analysis of fibrosis deposition showed a decrease in fibrosis levels in the tumor‐bearing mice group that almost reached the control levels (Figure 2B).” the level of fibrosis seems to be very different between the CTL and ATF3tg+cancer groups, and statistics are missing
Response: Sentence was edited accordingly and statistics was added.
Figure 3A: the number of mice used for cytometry must be indicated in the figure legend
Response: number of mice was added.
Figure 4: the quality of this figure is poor and there is a problem with the title
Response: Figure 4 was replaced by higher quality one.
Figure 4E: The CTL group is added in this qPCR analysis. Why only in this one and not in all the other qPCR analyses? and from which mice do these results come?
Response: Figure 4E is the only figure that the results are compared relative to control group. This was highlighted in the figure legend.
Minor points :
Line 24: text problem in the added sentence
Response: Corrected accordingly.
Line 123: tumours are not used in this article
Response: Corrected accordingly.
Line 144-155: font size different from the rest of the article
Response: Corrected accordingly.
Line 280-281 : I don't understand the meaning of this sentence? Comment made previously but no response. Can the authors specify what they mean by "peripheral symptoms"?
Response: This sentence was replaced by a more clarifying one.
The limitations paragraph must be moved up before the general conclusion
Response: Limitation section was moved accordingly.
Round 2
Reviewer 1 Report (Previous Reviewer 1)
In its present form, the revised manuscript is good enough for publication.
Author Response
We wish to thank the reviewer for finding our manuscript suitable for publication.
Reviewer 2 Report (Previous Reviewer 3)
The changes in this new version are substantial and clarify the message of the study. There are still a few minor changes to be made:
Line 37 and 329: spacing is not good
Line 148, 151, 154: degree "°" is not correctly noted
Line 285: infarction, not infraction
Line 309-310: it's the same sentence, the full stop is inappropriate
Author Response
We wish to thank the reviewer for the valuable comments.
All minor points were changed in the text.
This manuscript is a resubmission of an earlier submission. The following is a list of the peer review reports and author responses from that submission.
Round 1
Reviewer 1 Report
The present manuscript by Awwad and Aronheim analyzes the effect of tumor progression on cardiac function and remodeling in a mouse model of cardiomyocyte-specific overexpression of ATF3, a transcription factor known to be involved in cardiac hypertrophy, fibrosis and dysfunction. The manuscript is well written and most of the presented data are reasonable. However, there are a few points which are not clear yet and need to be addressed in the revised version of the manuscript. Thus, this reviewer recommends the paper for publication after some minor corrections:
11. The abstract is not pretty strong. It is more of an introduction to the topic than a summary of the results of this study. The data from the study should be briefly summarized in the abstract.
22. I´m not sure about the role of ATF3 in cardiac dysfunction. The authors claim that overexpressing ATF3 in cardiomyocytes induces hypertrophy and fibrosis, leading to cardiac dysfunction. However, in another publication using ATF3 knock-out mice, similar effects on cardiac function were observed when depleting ATF3 in combination with aortic banding (10.1371/journal.pone.0026744). This is contradictory and needs further explanation in the introduction.
33. The sentence in line 29 is misleading and sounds as if HF is induced by mouse models. Please reword the sentence to make it clear that the mouse models are used as a means to induce HF.
44. Line 38: Typing error “cancer cell proliferation”
55. When introducing M2 macrophages and their role in cardiac fibrosis in line 46, a study by Baumeier et al should be cited showing an association between high levels of M2 macrophages and lower levels of fibrotic tissue in inflammatory cardiomyopathy in humans (10.1007/s00395-020-00840-w).
66. Line 64: It is better to speak of an M1/M2 ratio rather than a shift from M1 to M2 macrophages, since it is not a shift from M1 to M2 but a different differentiation of monocytes.
77. Line 74: Please use the correct designation: C57BL/6.
88. Line 93: Please defined the abbreviation for LLC cell line.
99. Line 142: Typing error “45-60 min”
110. Line 153: “Liposomes were intraperitoneally (IP) injected…”
111. Please provide a reference for the statement in line 169, that expression of ATF3 in cardiomyocytes leads to hypertrophy, fibrosis and cardiac dysfunction.
112. The statement in line 186 about the reduction of ANP and BNP can only be made for ANP, since BNP is not statistically significant.
113. I was wondering about the number of mice used in Figure 1. Originally, n=7 (LLC injection) and n=8 (no injection) mice were used. However, in Figures 1C-G, only 5 and 6 animals were compared, respectively. Please provide an explanation for this discrepancy.
114. The conclusion from Figure 2B in line 205, that the tumor-group reaches fibrosis levels of the control group is not correct. A look on the data from Figure 2B shows a 5-fold higher level of fibrotic tissue in the “ATF3 tg + Cancer” group. Please also provide the p-value for this comparison in the figure.
115. Line 206: Please provide the abbreviation for TGFß and CTGF.
116. Figure 3A.: Please provide p-value for F4/80.
117. The macrophage markers used in Figure 3 are not unique. The use of single markers to determine macrophage types is critical because these markers are also expressed on other cell types. Quantification of double-positive cells, as performed in Figure 1A, should be applied more consistently. Please discuss this limitation of the study in the Discussion section.
118. Different font size in lines 275-280.
119. Line 320: typing error “5. Conclusion”
220. Line 321: “Using three different mouse models…” sounds like they were all used in this study. Please refer only to the one model that was used in this study.
The english language is basically good and needs only minor corrections.
Author Response
Reviewer #1
We wish thank the reviewer for the valuable comments.
- The abstract is not pretty strong. It is more of an introduction to the topic than a summary of the results of this study. The data from the study should be briefly summarized in the abstract.
Response: Abstract was corrected acccordingly.
- I´m not sure about the role of ATF3 in cardiac dysfunction. The authors claim that overexpressing ATF3 in cardiomyocytes induces hypertrophy and fibrosis, leading to cardiac dysfunction. However, in another publication using ATF3 knock-out mice, similar effects on cardiac function were observed when depleting ATF3 in combination with aortic banding (10.1371/journal.pone.0026744). This is contradictory and needs further explanation in the introduction.
Response: The debate regarding ATF3 role in hypertrophy was added in the discussion.
- The sentence in line 29 is misleading and sounds as if HF is induced by mouse models. Please reword the sentence to make it clear that the mouse models are used as a means to induce HF.
Response: sentence was changed to provide clarity.
- Line 38: Typing error “cancer cell proliferation”
Response: typo was corrected.
- When introducing M2 macrophages and their role in cardiac fibrosis in line 46, a study by Baumeier et al should be cited showing an association between high levels of M2 macrophages and lower levels of fibrotic tissue in inflammatory cardiomyopathy in humans (10.1007/s00395-020-00840-w).
Response: citation was added.
- Line 64: It is better to speak of an M1/M2 ratio rather than a shift from M1 to M2 macrophages, since it is not a shift from M1 to M2 but a different differentiation of monocytes.
Response: corrected accordingly.
- Line 74: Please use the correct designation: C57BL/6.
Response: Corrected.
- Line 93: Please defined the abbreviation for LLC cell line.
Response: LLC cell line was defined.
- Line 142: Typing error “45-60 min”
Response: Typing error was corrected.
- Line 153: “Liposomes were intraperitoneally (IP) injected…”
Response: The word “injected” was added.
- Please provide a reference for the statement in line 169, that expression of ATF3 in cardiomyocytes leads to hypertrophy, fibrosis and cardiac dysfunction.
Response: Reference was added.
- The statement in line 186 about the reduction of ANP and BNP can only be made for ANP, since BNP is not statistically significant.
Response: BNP is a hallmark of cardiac hypertrophy. The decrease in BNP mRNA levels was not statistically significant in tumor-bearing mice; however, a trend of decrease is shown. This was also emphasized in the text.
- I was wondering about the number of mice used in Figure 1. Originally, n=7 (LLC injection) and n=8 (no injection) mice were used. However, in Figures 1C-G, only 5 and 6 animals were compared, respectively. Please provide an explanation for this discrepancy. Response: Outliers were excluded using GraphPad prism, this was indicated in the methods section.
- The conclusion from Figure 2B in line 205, that the tumor-group reaches fibrosis levels of the control group is not correct. A look on the data from Figure 2B shows a 5-fold higher level of fibrotic tissue in the “ATF3 tg + Cancer” group. Please also provide the p-value for this comparison in the figure.
- Line 206: Please provide the abbreviation for TGFß and CTGF.
Repsonse: Abbreviation was added.
- Figure 3A.: Please provide p-value for F4/80.
- The macrophage markers used in Figure 3 are not unique. The use of single markers to determine macrophage types is critical because these markers are also expressed on other cell types. Quantification of double-positive cells, as performed in Figure 1A, should be applied more consistently. Please discuss this limitation of the study in the Discussion section.
Response: We agree with the reviewer about the limitation of using these macrophages markers, however, these markers are highly used in this area of research. This limitation was discussed.
- Different font size in lines 275-280.
Response: Font size was changed.
- Line 320: typing error “5. Conclusion”
Response: Typing error was corrected.
- Line 321: “Using three different mouse models…” sounds like they were all used in this study. Please refer only to the one model that was used in this study.
Response: The sentence was corrected accordingly.
Reviewer 2 Report
Main findings of the study:
The article written by Awward et al., provides a new breakthrough in the understanding of the incidence of tumor growth on heart. This study is of importance for the cardiooncology research field. These observations report for the first-time evidences of reduced cardiac hypertrophy and fibrosis level in a mouse model of ATF3 transgenic mice after lung tumor implantation. The author demonstrate that this amelioration of cardiac outcome is related to a macrophage pro-inflammatory toward immunomodulatory phenotype switch. Although the article is well written and the data brought by the authors are substantial, some missing controls have been identified and experimental questions have to be answered before publication.
Detailed review report:
Major concerns:
1. An experimental control consisting in ATF3 transgenic mice not receiving doxycycline is missing in most experimental sets. Although “Controls” are depicted in figures 2 A-B and 4 E-F these are missing in the other figures for unexpected reasons. Please explain what are those controls (wild type mice? ATF3 – dox mice?).
2. The authors have not specified the duration of the experiments and their timing in days and weeks. Some information on the kinetic are found in figures 1A and 4A but there I no detail about the duration of each step.
3. Please explain how mice were randomized before receiving cancer cell implantation.
4. The definitions of M1/M2 are antiquated. It is widely considered that there is a spectrum of phenotypes that exist in tissues and thus, M1/M2 are used for in vitro assessment only. Regarding this observation the author claim that there is a pro-inflammatory “M1” toward immunomodulatory “M2” macrophage switch ongoing ; however ; they don’t bring any evidence of the presence of pro-inflammatory “M1” macrophage in their model. Please asses the expression of pro-inflammatory macrophage.
5. A better characterization of macrophage population is required. Flow cytometry data tends to support an increased infiltration of F4/80+CD206+ immunomodulatory macrophages but F4/80+CD206- have not been characterized. Use more immunomodulatory markers to characterize the F4/80+CD206+ population and pro-inflammatory one to characterize the F4/80+CD206- population.
6. Chlodronate is well known to deplete all macrophage populations. Thus, it is difficult to interpreted and conclude about the relative role of immunomodulatory or pro-inflammatory macrophages.
Minor concerns:
1. Please Italicize Latin words such as in vivo and in vitro (e.g. abstract and p 4 line 26, p 12 line 28, p 14 line 12 etc…).
2. Overall main figures quality is very low.
3. Please modify rodent genes/proteins: Rodent genes/proteins must be written in lower case with the first letter capitalized and genes must be italicized (e.g. CCL2). Moreover, please refer to the official gene nomenclature for some genes that are not conform (e.g. Cd206 is encoded by Mrc1 gene).
Author Response
Reviewer #2
We wish thank the reviewer for the valuable comments.
- An experimental control consisting in ATF3 transgenic mice not receiving doxycycline is missing in most experimental sets. Although “Controls” are depicted in figures 2 A-B and 4 E-F these are missing in the other figures for unexpected reasons. Please explain what are those controls (wild type mice? ATF3 – dox mice?).
Response: This information was corrected in the methods section and controls that were used in each experiment have been defined clearly in the figure legend.
- The authors have not specified the duration of the experiments and their timing in days and weeks. Some information on the kinetic are found in figures 1A and 4A but there I no detail about the duration of each step.
Response: A more specified timeline of each experiment have been added to the figures.
- Please explain how mice were randomized before receiving cancer cell implantation.
Response: After genotyping, transgenic mice were randomly weaned into separate experimental cages.
- The definitions of M1/M2 are antiquated. It is widely considered that there is a spectrum of phenotypes that exist in tissues and thus, M1/M2 are used for in vitro assessment only. Regarding this observation the author claim that there is a pro-inflammatory “M1” toward immunomodulatory “M2” macrophage switch ongoing ; however ; they don’t bring any evidence of the presence of pro-inflammatory “M1” macrophage in their model. Please asses the expression of pro-inflammatory macrophage.
Response: TNFa is used as a marker for M1 macrophages. - A better characterization of macrophage population is required. Flow cytometry data tends to support an increased infiltration of F4/80+CD206+ immunomodulatory macrophages but F4/80+CD206- have not been characterized. Use more immunomodulatory markers to characterize the F4/80+CD206+ population and pro-inflammatory one to characterize the F4/80+CD206- population.
Response: F4/80+CD206- macrophages are shown in the gating strategy in the supplemental material (Figure 2, 3).No differences observed in these macrophages.
- Chlodronate is well known to deplete all macrophage populations. Thus, it is difficult to interpreted and conclude about the relative role of immunomodulatory or pro-inflammatory macrophages.
Response: We agree with the reviewer’s comment, by depleting macrophages using Clodronate liposomes, we have concluded that macrophages in general, play a crucial role in mediating this Heart-tumor cross-talk. We strongly believe that this study is the first step and that a more comprehensive study about the involvement of specific macrophages subtypes needs to be conducted.
Minor concerns:
- Please Italicize Latin words such as in vivo and in vitro (e.g. abstract and p 4 line 26, p 12 line 28, p 14 line 12 etc…).
Response: Corrected in text.
- Overall main figures quality is very low.
Response: Figures quality was edited and made clearer.
- Please modify rodent genes/proteins: Rodent genes/proteins must be written in lower case with the first letter capitalized and genes must be italicized (e.g. CCL2). Moreover, please refer to the official gene nomenclature for some genes that are not conform (e.g. Cd206 is encoded by Mrc1 gene).
Reviewer 3 Report
In this study, the authors investigated the role of tumour progression on cardiac dysfunction genetically induced by the expression of human ATF3 in mouse cardiomyocytes. The authors focused in particular on the processes of fractional shortening, hypertrophy and fibrosis. They conclude that macrophages play a role in this process. This is an interesting idea, which could provide a better understanding of the complex interactions between cancer and CVD.
Specific comments:
A number of details are missing from the materials and methods, making the results difficult to understand.
- Which mice were used? male or female?
- line 82: "while single transgenic were used as controls" no data in this article are described in these mice
- The Figure 1 protocol is illegible and incomplete. No time data are given, making it difficult to understand the study protocol. Furthermore, it does not clearly indicate the control group which, according to the figures, is the cancer-free group (and not the single transgenic lines). The problem is much the same in figure 4, although the text is more legible.
- The number of mice used in the experiments is not at all clear. Sometimes, but not always, the number of animals studied is indicated in the legend. It would be clearer to indicate the number of mice tested in each experiment directly on the protocols shown in Fig. 1 and 4. Furthermore, if we look at the number of mice tested in the results, 5-6 were used for qPCR, 4 for labelling and an unspecified number for cytometry. However, ultrasound scans were only performed on 7-8 mice: why were not all the mice monitored functionally? and if so, how were the mice chosen in each group? This is important because some of the results show a wide spread.
Concerning the results:
-Line 176: and "cardiac function was not significantly deteriorated upon Dox addition (Figure 1B and Supp. Figure 1A)". This conclusion is not clearly illustrated in these 2 figures. There are 5 mice in Suppl Fig 1A and 7 in Fig 1B: why was the follow-up not carried out on the same mice? A clearly illustrated longitudinal follow-up would be more convincing.
-Line 178: "In contrast, tumor-bearing mice, that showed decreased FS levels prior to cancer cells injection", this result is not shown anywhere.
-Line 179-180: "In the latter, FS reached cardiac contractile levels that are almost similar to healthy mouse (30%, Figure 1B-C)." Which mice are healthy? Where is this result?
- Figure 1E: How many hearts were tested and how many sections per heart? In the absence of quantification, it would be more convincing to have the whole sections and the indication on this whole section of the magnifications shown in the figure.
-Line 188: “This reduced hy‐ 186 pertrophy in tumor‐bearing mice was also accompanied by the reduction in hypertrophic 187 hallmark gene markers ANP and BNP (Figure 1F‐G)”. The reduction in BNP was not significant.
Line 203-204: "Consistently, quantitative analysis of fibrosis deposition showed a decrease in fibrosis levels in the tumor-bearing mice group that almost reached the control levels (Figure 2B)". Were statistics generated for the control group? Which mice are in this control group? As in Figure 1, how many slices per heart were tested? A single heart slice is not enough to reach a conclusion.
Figure 4: macrophage depletion. As with the other figures, the graphs should be switched to individual values with averages and SEM for a better appreciation of the results. In this figure, a control group appears: what is it? Why was it not used in the previous experiments?
Regarding the discussion:
-Line 272-273: "Drug therapies to treat CVD and HF typically address pe- 272 ripheral symptoms without directly targeting the molecular defects within the heart.". This sentence needs to be qualified because anticalciques and other inhibitors of ion channels or regulators of these channels are very widely used to treat CVD and act directly on intracellular cardiac targets.
Concerning supplemental data:
Without detailed legends, these figures are difficult to understand. In addition, the cytometry data are illegible, making it impossible to understand what has been done.
The tables show the results of the FS calculations, but only the averages are shown. The SDs must be added.
Minor comment:
Figure 3: "B" and "C" are no longer on the corresponding graphs.
Author Response
Reviewer #3
We wish thank the reviewer for the valuable comments.
In this study, the authors investigated the role of tumour progression on cardiac dysfunction genetically induced by the expression of human ATF3 in mouse cardiomyocytes. The authors focused in particular on the processes of fractional shortening, hypertrophy and fibrosis. They conclude that macrophages play a role in this process. This is an interesting idea, which could provide a better understanding of the complex interactions between cancer and CVD.
Specific comments:
- Which mice were used? male or female?
Response: All experiments were conducted on male mice, this data is indicated in the figure legends and in the methods section.
- line 82: "while single transgenic were used as controls" no data in this article are described in these mice.
Response: We have defined the controls of each experiment in the legend of each figure.
- The Figure 1 protocol is illegible and incomplete. No time data are given, making it difficult to understand the study protocol. Furthermore, it does not clearly indicate the control group which, according to the figures, is the cancer-free group (and not the single transgenic lines). The problem is much the same in figure 4, although the text is more legible.
Response: Timeline was edited. Time data was added. Control groups have been defined clearly.
- The number of mice used in the experiments is not at all clear. Sometimes, but not always, the number of animals studied is indicated in the legend. It would be clearer to indicate the number of mice tested in each experiment directly on the protocols shown in Fig. 1 and 4. Furthermore, if we look at the number of mice tested in the results, 5-6 were used for qPCR, 4 for labelling and an unspecified number for cytometry. However, ultrasound scans were only performed on 7-8 mice: why were not all the mice monitored functionally? and if so, how were the mice chosen in each group? This is important because some of the results show a wide spread.
Response: Number of mice is indicated in each figure legend. Outliers were excluded using GraphPad Prism, this was indicated in the methods section.
- Line 176: and "cardiac function was not significantly deteriorated upon Dox addition (Figure 1B and Supp. Figure 1A)". This conclusion is not clearly illustrated in these 2 figures. There are 5 mice in Suppl Fig 1A and 7 in Fig 1B: why was the follow-up not carried out on the same mice? A clearly illustrated longitudinal follow-up would be more convincing.
Response: This is a different experiment in which we examined how Dox addition affects the cardiac dysfunction caused by the ATF3 expression. Timeline of this experiment was added to supplemental figure 1.
- Line 178: "In contrast, tumor-bearing mice, that showed decreased FS levels prior to cancer cells injection", this result is not shown anywhere.
Response: This result is shown in figure 1C and was emphasized in the text.
- Line 179-180: "In the latter, FS reached cardiac contractile levels that are almost similar to healthy mouse (30%, Figure 1B-C)." Which mice are healthy? Where is this result?
Response: A reference of healthy mouse fractional shortening was added accordingly.
- Figure 1E: How many hearts were tested and how many sections per heart? In the absence of quantification, it would be more convincing to have the whole sections and the indication on this whole section of the magnifications shown in the figure.
Response: 4 hearts were tested, 5 sections per heart. This information was added to the figure legend.
- Line 188: “This reduced hy‐ 186 pertrophy in tumor‐bearing mice was also accompanied by the reduction in hypertrophic 187 hallmark gene markers ANP and BNP (Figure 1F‐G)”. The reduction in BNP was not significant.
Response: BNP is not statistically significant. However, an improvement trend is observed in tumor-bearing mice. The text was edited accordingly.